# Research on Time-Aware Group Query Method with Exclusion Keywords

**Liping Zhang, Jing Li and Song Li \***

School of Computer Science and Technology, Harbin University of Science and Technology, Harbin 150080, China; zhangliping0730@hrbust.edu.cn (L.Z.); 1704010306@stu.hrbust.edu.cn (J.L.)
\* Correspondence: lisongbeifen@hrbust.edu.cn

**Abstract:** Aiming at the problem that the existing spatial keyword group query problem did not consider the query requirements with exclusion keywords and time attributes, a time-aware group query problem with exclusion keywords (TEGSKQ) is proposed for the first time. To solve this problem effectively, this paper proposes a query method based on the EKTIR-Tree index and dominating group (EKTDG). This method first proposes the EKTIR-tree index, which incorporates Huffman coding and integrates Bloom filters to deal with excluded keywords in order to improve the hit rate of keyword queries, significantly improving the query efficiency and reducing the storage occupancy. Then, the Candidate algorithm is proposed based on the EKTIR-tree index to filter out the spatial–textual objects that meet the query's keywords and time requirements, narrowing the search space for subsequent queries on a large scale. To address the problem of the low efficiency of existing algorithms based on a spatial distance query, a distance-dominating group is defined and a pruning algorithm based on a spatial distance-dominating group is proposed, which is a refining process of query results and greatly improves the search efficiency of the query. Theoretical and experimental studies show that the proposed method can better handle group queries with exclusion keywords based on time awareness.

**Keywords:** time-aware; group query; exclusion keywords; spatial keyword query; Huffman coding

## 1. Introduction

With the rapid development of Internet technology and sensor technology in recent years, location-based services have been commonly used in daily life. In particular, spatial keyword query, as one of the important technologies in location-based services, has been widely used in many fields, such as intelligent navigation systems and spatial positioning systems. Different types of spatial keyword query problems have been studied in depth by scholars at home and abroad, such as spatial keyword nearest neighbor query problems [1–3], TOP-k spatial keyword queries [4–8], inverse nearest neighbor queries [9–12] and spatial keyword group queries [13–15]. In the spatial database, a number of spatial–text objects, also known as points of interest (POI), are stored. A spatial keyword query returns one or some points of interest, which need to meet the various requirements of the query and be the best distance. Among the various branches of spatial keyword queries, the application of spatial keyword group queries in daily life has gradually spread in recent years and attracted the attention of many scholars. For example, when a tourist travels to a certain city, planning the desired location of the hotel needs to include activities such as eating, visiting the park, watching movies and so on for a period of time.

Temporal information is also an indispensable consideration in the field of spatial keyword queries. Since each POI such as stores or parks has its own opening hours, users cannot access the POI during non-opening hours. In the query system or recommendation system, different POIs have different opening hours, for example, the breakfast store is open from 6:30 to 10:00 and the shopping mall is open from 9:30 to 20:00, so for the different

needs of different users, it is necessary to return different POI objects. Specifically, when the user wants to eat breakfast at 7:00, the query system should return the above breakfast store instead of the mall. Although there may be restaurants in the mall, the mall's time does not meet the user's needs. However, most researchers have focused on textual constraints and spatial constraints and have not considered the impact of temporal constraints in daily life.

Furthermore, with the development of science and technology in society, the needs of users are also increasing. For spatial keyword queries, users may not only want to query to meet the required keyword objects, but may also hope that, in the query process, they can be excluded from containing the user's exclusion keyword objects. Moreover, some users around the world also have many exclusionary matters due to their religious beliefs and other reasons. However, few studies on spatial keyword querying have considered users' exclusion preferences. Therefore, spatial keyword querying with exclusion keywords has great research value.

Currently, the existing spatial keyword group queries only have keyword information and distance constraints and do not consider both temporal information and exclusion keywords. Therefore, this paper proposes a time-aware group query method with exclusion keywords. To deal with this problem effectively, this paper proposes a query method based on the EKTIR-Tree index and dominating group which is called EKTDG. The main contributions of this paper are as follows:

(1) Aiming at the situation where traditional spatial keyword group query research cannot handle both temporal properties and exclude keyword information, this paper proposes a new query model, namely temporally aware group query with exclusion keywords. This query model limits both temporal information and exclusion keyword information based on traditional spatial keyword group queries, which is more suitable for the multiple query needs of users in current society.

(2) In response to the problem that the current existing indexing techniques cannot handle the query model proposed in this paper, a new index, the EKTIR-tree, is proposed in this paper. The index not only has the advantages of the IR-tree, but also introduces the idea of Huffman coding to improve the query efficiency. In handling the exclusion keywords, the index introduces the Bloom filter, which can handle the exclusion keyword information efficiently. To improve the query efficiency, a pruning query algorithm called the Candidate algorithm is further proposed based on the proposed EKTIR-tree index. This algorithm uses the EKTIR-tree index to perform the first step of pruning operation to derive the objects in the spatial–text database that meet the query requirements of time constraints and keyword constraints, thus reducing the computational overhead of subsequent queries.

(3) In order to solve the problem of the low efficiency of the traditional spatial keyword group query algorithm, this paper further proposes a query algorithm based on a distance-dominant group. This algorithm first selects the dominant objects of the distance-dominant group to initially reduce the search space and then further reduces the search space according to the relationship between the dominant objects to effectively improve the query performance and efficiency.

Compared with the previous methods, CD-Exact, the operation efficiency of the proposed method is improved by at least 14%. The rest of the paper is organized as follows. Section 3 of this paper gives the relevant important definitions. Section 4.1 proposes a new index, the EKTIR-tree index, and Section 4.2 proposes a filtering algorithm based on this index. Section 5 of this paper proposes a spatial keyword group query method based on the distance domination group. Section 6 gives the corresponding experimental analysis and Section 7 gives a summary. The method proposed in this paper is 14% more efficient than the previous method. The algorithm relationship is shown in Figure 1.

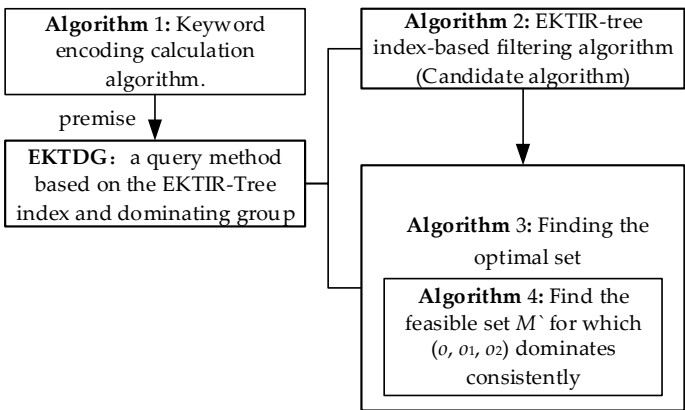

**Figure 1.** The figure of algorithm relationship.

## 2. Related Work

The spatial keyword group query problem [13–15], as a special form of spatial keyword query, has common applications in real life. The concept of the mCK (m-closest keywords) query was first proposed in the literature [16]. Given a set of query keywords, the mCK query finds m interest points and the keyword information of this set of interest points jointly covers the query keywords and minimizes the maximum pairwise distance of the objects in the group. However, Zhang et al. [16] only focused on the fact that each object contains only one query keyword and its proposed exact algorithm has low adaptability for large data sets. Considering the possibility of multiple keywords, Guo et al. [17] relaxed the constraint of having only one keyword per object and proposed an approximation algorithm with a factor of $(1.15 + \varepsilon)$. Deng et al. [18] presented an optimal keyword coverage problem, which is a variant of the mCK query. This study considered both intra-group distance and keyword weights and incorporated both factors into a linear cost function, thereby proposing an accurate algorithm to solve this problem. Since the results of mCK queries do not provide sufficient support in some application scenarios, a new spatial keyword query, called the SK-Cover query (spatial keyword cover query), was proposed in the literature [19]. This query considered the number of objects in the results set and put an approximate algorithm with a time complexity of $O(\log m)$ for this query problem. Effective access policies and pruning rules were laid out to improve the efficiency and scalability of the algorithm in [19]. Subsequently, Li et al. [20] studied spatial keyword group queries and proposed a parametric approximation algorithm that allows the approximation ratio to be adaptive and the user to assign arbitrary query precision.

The literature [13–20] are traditional spatial keyword group queries, focusing only on the study of keyword constraints and distance constraints. Nevertheless, some researchers have extended conventional spatial keyword group queries to some extent. Singh et al. [21] considered objects labeled with keywords and embedded in a vector space to study the multi-objective query problem in a multi-dimensional space for the closest set of points satisfying a given set of keywords. In a low-dimensional space, IR trees are mainly used to support query processing. But in a high-dimensional space (with dimensions larger than 10), different MBRs (minimum bounding rectangles) of IR trees can contain a large amount of overlapping data, resulting in a reduced index performance. So, this literature studied indexing mechanisms and query algorithms in a high-dimensional space, using random projection and hash-based indexing structures, and achieved high scalability and speedup ratios. The CoSKQ (collective spatial keyword query) query model introduces query points based on mCK, and Gao, Cao and Zhao et al. [22–24] studied the CoSKQ query model in a road network space. Unlike most studies, Zhao et al. [24] proposed a popularity-aware aggregated keyword in road networks, aiming to find a set of popular POIs (i.e., a popularity region). The POIs cover the query keywords and satisfy the distance requirements from each node to the query node and between each node pair, such that the sum of the scores of these nodes for the query keywords is maximized. For this reason,

a scaling technique for scoring was raised to reduce the search space, and a redundant computation reduction technique was proposed to reduce the redundant computations in query processing. Su et al. [25] also studied ensemble spatial keyword queries in a road network environment, but the difference is that it is based on group-based ensemble queries, i.e., GBCK (group-based collective keyword). Considering the importance of the keyword levels for decision support, Zhang et al. [26] proposed the Level Aware Set Space Keyword Query (LCSK) to find a set of POIs that jointly covers the query keywords with threshold constraints and minimal spatial distance cost. An exact algorithm and approximation algorithm with provable approximations related to this problem were designed for the LCSK. To better express more fine-grained preferences for cost-aware and distance-constrained keyword queries in the set space, Chan et al. [27] proposed new criteria, optimized the cost function and designed an exact and approximate algorithm. Xu et al. [28] differed from other works in that it studied mobile set space keyword queries from the dynamics of query points and raised two approximation algorithms based on the safety zone technique.

Considering the importance of temporal information in spatial keyword queries, some researchers have proposed query models with temporal awareness. Chen et al. [29] first studied temporally aware Boolean keyword queries, known as a TABSKQ (time-aware Boolean spatial keyword query), and proposed an efficient index structure TA-tree and its corresponding query algorithm for this problem model. The index can effectively prune the search space and take into account both keyword information and temporal information. Chen et al. [30] proposed two evaluation functions for time-aware aggregate keyword queries to meet different types of query requirements and both proposed corresponding processing algorithms. Chan et al. [31] studied the model of indoor keyword routing with time constraints, i.e., the TIKRQ problem (time-constrained indoor keyword-aware routing query), and proposed a series of pruning rules and corresponding solution algorithms. Considering that most of the existing works cannot solve the spelling error cases and thus focus on the time-aware approximate set keyword search in traffic networks, Feng et al. [32] proposed a TDAG-tree index for the distance pruning of query objects and designed two approximation algorithms to improve the processing efficiency to a large extent.

## 3. Definitions and Symbol Descriptions

According to the content of the research and related technologies applied, this section provides the following basic definitions.

In this paper, a spatial–text database $D = \{o_1, o_2, \ldots, o_n\}$ is given. The object $o_i$ in $D$ is denoted as a tuple ($o_i.loc$, $o_i.K$, $o_i.st$, $o_i.et$), each POI contains its own spatial coordinates $o_i.loc$, keyword set $o_i.K$, start time $o_i.st$ and end time $o_i.et$, where the set of keywords $o_i.K = \{key_1, key_2, \ldots\}$.

**Definition 1.** *TEGSKQ. A time-aware spatial keyword group query with exclusion keywords (TEGSKQ) is denoted as q = (q.loc, q.K+, q.K−, q.st, q.et), where q.loc is the location coordinate of the query point in the Euclidean space, q.K+ is the set of positive keywords of the query, representing the user's preference, q.K− is the set of repulsive keywords of the query, representing the user's repulsive intention, q.st is the starting time point of the query-specified time period and q.et is the end time point of the query-specified time period. The query q should return the feasible set with the optimal distance. It is to make the objects in the feasible set as compact as possible and closest to the query point as possible. The concept of the feasible set is shown in Definition 2.*

For example, a pair of friends plan to visit an area and expect to leave from the hotel where they are staying at 9:00 to return to the hotel at 20:00. The pair suggest that they would like to eat food such as pizza or fried rice and suggest excursions such as amusement parks and parks, malls with cafes, etc. But they have repulsive intentions, such as being repulsive to restaurants with pork dishes and disliking cafes with music. At this point, the demand proposed by the pair of friends conforms to the TEGSKQ query model, $q = (q.loc$:

the hotel location, *q.K+*: {pizza, fried rice, amusement parks, parks}, *q.K−*: {pork, cafe with music}, *q.st*: 9:00, *q.et*: 20:00).

**Definition 2.** *Feasible set M. Given a TEGSKQ query denoted as q and let M be a feasible set of q, then M satisfies the following conditions:*

(1)   *$q.K+ \subseteq \cup o_i.K_{o_i \in M}$, the keyword concatenation of objects in M can cover all the keywords in q.K+.*

(2)   *$\forall key_{key \in q.K-} \notin \cup o_i.K_{o_i \in M}$, no object in M can contain any of the exclusion keywords specified by query q.*

(3)   *$\forall o_i.(st,et)_{o_i \in M} \subseteq q.(st,et)$, the time zone of any object in M must be included in the time zone of q.*

**Definition 3.** *The diameter of the feasible set M.Dia. Given a feasible set M, when there is only one object in M, the diameter of the feasible solution set M.Dia is 0. When the feasible set contains a set of objects, the diameter of the feasible set M.Dia is the diameter of its enveloping circle, i.e., $M.Dia = \max\limits_{o_i,o_j \in M} dist(o_i, o_j)$. $dist(o_i, o_j)$, which is the Euclidean distance between the two points.*
*M.Dia quantifies the compactness of the objects in the feasible solution set; the more compact, the closer the query point is to each object.*

**Definition 4.** *The distance between the feasible set and query point Dist(q, M). Given a TEGSKQ with its feasible set M, the distance between the feasible set M and the query point q, Dist(q, M), is the distance between the query point and the object in the feasible set that is farthest away from the query point, $Dist(q, M) = \max\limits_{o \in M} dist(q, o)$*

**Definition 5.** *The spatial distance cost of a feasible set M.Scost. Given a feasible solution set M, where the spatial distance cost is denoted as M.Scost, the formula is calculated as shown in Equation (1):*

$$M.S\cos t = \alpha \times M.\text{dia} + (1 - \alpha) \times Dist(q, M) \tag{1}$$

The smaller the value of *M.Scost*, the greater the possibility that its corresponding feasible set *M* becomes the final result set of a TEGSKQ. Here, a linear approach is used to calculate the distance cost, which can be more intuitive to understand the magnitude of the weights of both. $\alpha$ is a smoothing parameter to balance the compactness between objects in the feasible set and the distance between the feasible set and the query point. In this paper, for the convenience of the study, taking $\alpha = 0.5$, the distance cost of the feasible set can be simplified to Equation (2):

$$M.S\cos t = M.\text{dia} + Dist(q, M) \tag{2}$$

**Definition 6.** *The time overlap η [29]. Given a TEGSKQ denoted as q with a spatial–text object o, where such that t = {st, end}. η(q, o) can be calculated according to Equation (3):*

$$\eta(q, o) = \frac{|q.t \cap o.t|}{|q.t|} \tag{3}$$

$\eta(q, o)$ is the temporal overlap between query *q* and object *o*. Quantitatively, it represents the temporal fit between object *o* and query *q*. The larger the value, the higher the probability that object *o* meets the temporal requirements of the query.

## 4. Pruning Filtering Method Based on the EKTIR-Tree Index

The traditional spatial keyword group query can no longer meet the diverse query needs of today's society. In addition to the keyword constraints, the user may also have limited requirements for the time and will put forward their own exclusion intentions in exclusion keywords as one of the query filtering conditions. For such queries, this

paper first constructs a new index, the EKTIR-tree, for spatial–text database *D* and then performs pruning queries based on the EKTIR-tree in spatial–text database *D* to select all candidate objects that meet the positive keywords, exclusion keywords and time constraints. The whole process can be split into two parts: index construction and index-based pruning query.

### 4.1. EKTIR-Tree Index

This section proposes a new index to handle spatial keyword group queries with time-constrained exclusion keywords. Firstly, the relevant definitions are proposed, as shown in Definitions 7–9.

**Definition 7.** *The keyword documentation doc(·). Given region A, there is a set of spatial–text objects o in the region and the keyword information of all objects is collected into a document, which is the keyword document of the current region, denoted as* $doc(A) = \underset{o \in A}{\cup} o.key$.

**Definition 8.** *The keyword weight w. Given a spatial–text database D containing many spatial–text objects o, the keywords of all o in D are put into a set to form a document. The weight of each keyword in the document is calculated using the TF-IDF [33] method in the field of natural language processing, which is subsequently used to sort the keywords and construct Huffman trees and keyword encoding.*

**Definition 9.** *The keyword encoding* $Code_{key}$. *The keywords of all spatial–text objects o in D are constructed into Huffman trees according to their respective word frequencies f and then the Huffman encoding* $Code_{key}$ *of each keyword is obtained.*

Since the R-tree series index can reduce the query space range efficiently, this paper proposes an EKTIR-tree index based on an IR-tree. An example of POI distribution is given, as shown in Figure 2.

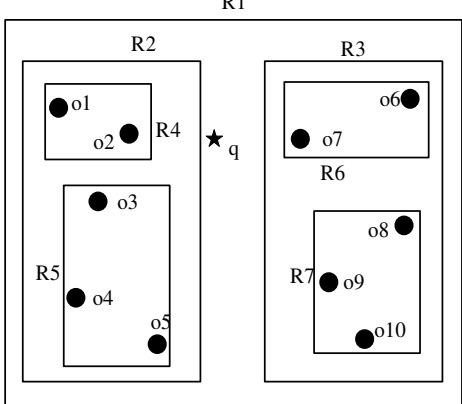

**Figure 2.** The figure of POI's distribution.

Figure 2 shows a batch of POI objects distributed in the spatial–text database D. Firstly, these POIs are spatially divided according to the division rules of R-tree, followed by the construction of the EKTIR-tree index, as shown in Figure 3.

The format of the non-leaf node of the EKTIR-tree index is (cp, MBR, info), where cp is the pointer of this node to its children, MBR is the minimum outer wrapping rectangle of its children pointed by cp and info is the information file of this node. And the internal format of info is (InvFile, BloomFilter, Ukey, Utime), where InvFile is the inverted file formed by the special processing of all POI keyword information in the node, BloomFilter is the Bloom filter formed by the keyword document doc (R*i*) for the node and Ukey is the intersection of the Ukey content of the node's children, built from the bottom up. Ukey is used to

collaborate with BloomFilter to filter the excluded keywords. Utime is the concatenation of the time intervals of all POIs in the node. The leaf node of the index contains a set of POIs in the region of the node; in addition, each leaf node also contains a pointer to the information file (info) of the node. The format of the info file is (CodeArr, BloomFilter, Utime), where CodeArr is the keyword encoding of all POIs in the leaf node in order and the BloomFilter is similar to Utime and the non-leaf nodes.

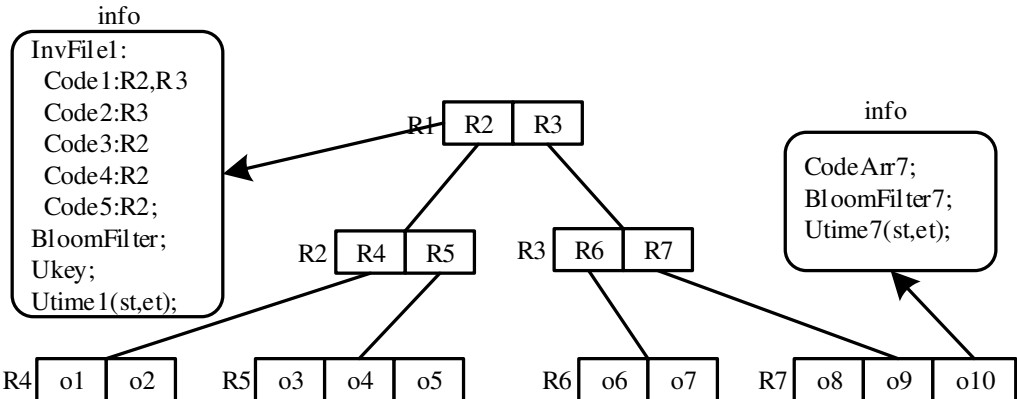

**Figure 3.** The EKTIR-tree index.

The non-leaf node of this index adds a time attribute and a filter to handle the exclusion keyword to the IR-Tree and specializes the inverted files in the IR-Tree. The format of each record in a traditional inverted file is (key: document1, document2 . . .), which takes each keyword key as an index item and the object containing the keyword as the index value. However, when the EKTIR-tree generates the inverted file, each keyword is arranged in descending order by weight in advance, according to the weight size of the desired keyword in the file. And the index item is no longer the keyword itself when stored, but the keyword's corresponding Huffman-coded $Code_{key}$ is used. As a result, each record in the EKTIR-tree's inverted file has the format ($Code_{key}$:R1, R2, . . .), which has the advantage of being processed as follows: First, the index entries of each record are arranged according to weights, allowing for the faster finding of the target record when looking for data in the inverted file. Secondly, each index item adopts the form of $Code_{key}$ instead of keywords, which can reduce the space storage consumption to a certain extent.

Since the number of POIs in the leaf nodes is not as many as in non-leaf nodes, the inverted indexing strategy of using the IR-tree in leaf nodes occupies more space. To reduce storage space, the leaf nodes of the EKTIR-tree index arrange the encoding of keywords in the order of weight to form a string of bits, which can use the efficient KMP algorithm when matching keywords. One of the main features of Huffman coding is non-prefix matching coding, which allows individual codes to be concatenated and decoded without adding any interval symbols in between. Moreover, Huffman coding is the code with the shortest average code length, so this strategy not only reduces space storage consumption, but also does not have a large speed reduction in matching keywords.

When the text or time information of a POI changes, the EKTIR-tree index will first locate the leaf node where the POI is located, check the information and update the extended information of the leaf node and then update the extended information of the parent node iteratively, according to the change information, until the upward parent node does not need to update.

Since the use of Huffman coding can significantly improve the query efficiency of indexes and greatly reduce the space occupation, in order to improve the efficiency and quality of index construction, this paper gives Algorithm 1 to calculate the keyword coding.

---

**Algorithm 1:** Keyword encoding calculation algorithm

---

**Input**: Spatial–text database $D$.
**Output**: doc($D$) corresponds to the code set Code$\{Code_{k1}, Code_{k2}, \ldots\}$.
begin
1:  doc($D$) $\leftarrow \cup o.key$; $HT \leftarrow \varnothing$; /* Get all keywords and declare an empty tree node */
2:  $W: \{ w_{k1}, w_{k2}, \ldots\} \leftarrow$ TF-IDF(doc($D$)); /* Calculate the weight of each keyword using the TF-IDF model */
3:  TreeSet $T$.add($W$); /* Put the weight data into a TreeSet object $T$ and sort it by increasing weight value */
4:  while $T$ is not empty then /* Constructing a Huffman tree $HT$ */
5:  Take the two keywords $key_n$, $key_{n-1}$ with the smallest weight as the left and right nodes, and generate a virtual tree node $f$ with $w_f = w_n + w_{n-1}$, and node $f$ is the parent node of the keywords $key_n$, $key_{n-1}$;
6:  $T$.add($w_f$);
7:  end while
8:  $HT \leftarrow$ The final generated virtual node;
9:  Code$\{Code_{k1}, Code_{k2}, \ldots\} \leftarrow$ Get the code of each keyword from $HT$;
10: return Code$\{Code_{k1}, Code_{k2}, \ldots\}$;
end

---

The Huffman encoding of each keyword is obtained by Algorithm 1. The smaller the length of the encoding, the more frequently the keyword appears. In the construction of the EKTIR-tree, this encoding is involved in the construction of the CodeArr of the leaf nodes and the formation of the inverted file of the non-leaf nodes. For example, suppose the keyword set of a leaf node is {$key1$, $key3$, $key5$} and the calculated Code is {$Code_{key1} = 0$, $Code_{key3} = 11$, $Code_{key5} = 100$}, then the CodeArr in the leaf node information file is 011100. Furthermore, suppose the keyword set of a non-leaf node is {$key1$, $key2$, $key3$, $key4$, $key5$} and the calculated Code is {$Code_{key1} = 0$, $Code_{key2} = 101$, $Code_{key3} = 11$, $Code_{key4} = 1000$, $Code_{key5} = 100$}, then the inverted file in the information file of the node part of the format is {$Code_{key1}: \ldots$; $Code_{key3}: \ldots$; $Code_{key2}: \ldots$}. The order of each record is sorted by the length of the index keyword code in ascending order, meaning that the keyword with the highest weight is in the front of the reverse file record, thus improving the efficiency of each search.

Suppose that the total number of POI keywords in this database $D$ is $k$, that is, the number of leaf nodes of the Huffman tree is $k$, the time complexity of sorting and other operations using the Treeset data structure is O($\log k$) and the time complexity of building the Huffman tree and obtaining the Huffman code is O($k \log k$). This means that the time complexity of Algorithm 1 is O($k \log k$).

*4.2. Pruning Query Method Based on EKTIR-Tree Index*

Following the construction of the EKTIR-tree index, this subsection performs keyword and temporal pruning based on the index. It searches for all POI objects that meet the keyword and exclusion keywords and temporal requirements of query q and scores each object into a hash table to facilitate subsequent processing. For more efficient pruning, Theorems 1–3 are first given:

**Theorem 1.** *Given a spatial–text database D and the corresponding EKTIR-tree index and query q, note that a non-leaf node in the index is N. If its BloomFilter determines that the keyword key does not appear in q.K−, then there is absolutely no exclusion keyword in the children nodes of N.*

**Proof of Theorem 1.** According to the working principle of the Bloom filter, assuming the existence of the exclusion keyword key in the children nodes of $N$, there must be key $\in$ doc($N$). When constructing the Bloom filter, the bit of key after hash function mapping must be 1 instead of 0. Therefore, after verifying, the exclusion keyword key of $q$ by the Bloom filter is calculated to exist. Contrary to the original condition, the proof is complete. □

**Theorem 2.** *Given a spatial–text database D and the corresponding EKTIR-tree index and query q, note that a non-leaf node in the index is N. If its BloomFilter determines that the keyword key appears in the exclusion keyword set q.K−, the result is that it exists. And the keyword intersection of non-leaf node N contains a keyword key in the set of query exclusion keywords, i.e., $\underset{key \in q.K-}{\exists}$ key ∈ N.Ukey. Then, the key exists in the Ukey of all children nodes of this node and this node and its descendant nodes should all be pruned.*

**Proof of Theorem 2.** Using the converse method, assume that *key* ∉ ∀$N_i$.*Ukey* because *N.Ukey* = ∩$N_i$.*Ukey*, then *key* ∉ *N.Ukey* contradicts the original condition and the proof is complete. □

**Theorem 3.** *Given a query q and an EKTIR-tree index, if the time zone of the query does not have any intersection with the time zone of the current node, i.e., q.st > N.info.et or q.et < N.info.st, then the node and its descendants can be pruned in full.*

**Proof of Theorem 3.** Using the converse method, assume that there exists a POI object $o_i$ in the MBR region of the node that intersects with the time interval of the query *q*, i.e., $o_i.t \cap q.t \neq \varnothing$. Since the Utime in the information file of each node in the EKTIR-tree is a concatenation of the Utimes of its child nodes, then $o_i.t \subseteq N.Utime$ and $q.t \cap N.Utime \neq \varnothing$, contradicting the original condition, and the proof is complete. □

The scoring function for POI is further given in Equation (4):

$$Score(o) = \beta \cdot \frac{1}{dist(o,q)} + (1-\beta) \cdot \eta(o,q) \tag{4}$$

The fractional values of the non-leaf nodes are calculated by Equation (5):

$$Score(N) = \beta \cdot \frac{1}{dist(o_i,q)} + (1-\beta) \cdot \eta(N,q) \tag{5}$$

In Equation (5), $o_i$ is the nearest child node among the child nodes to the query point and $\beta$ is the smoothing parameter to balance the spatial distance and temporal overlap degree. The closer to the query point or the higher the overlap between time and query, the larger the Score value. Without losing generality, this article takes $\beta$ as 0.5.

Specifically, take the R4 node in Figure 2 as an example and assume that the Euclidean distance between $o_1$ and $q$ in R4 is 4, the time of $o_1$ is (9:30, 20:30), the Euclidean distance between $o_2$ and $q$ is 2, and the time of $o_2$ is (7:00, 11:00) and the time of $q$ is (9:00, 21:00), then the scoring is calculated for $o_2$ to get Score($o_2$) = 1/2 + 4/12 ≈ 0.83 and for non-leaf nodes R4 to get Score(R4) = 1/4 + 11/12 ≈ 1.167.

Based on the EKTIR-tree index, combined with Theorems 1–3, the traversal starts from the root node and determines whether the query requirements are met in terms of time and keywords, respectively. A priority queue is used to save the intermediate results until the POI is traversed and the candidate POI is computed as a Score value into a hash table, further giving Algorithm 2.

Algorithm 2 first initializes a priority queue $U$ to store the nodes that need to be recorded during the query and a node is queued when queue $U$ is not empty (lines 1–4). If the currently traversed node is a POI and matches the query requirement in terms of keywords and time, its spatial distance cost is calculated according to Equation (4) and stored in the resultant hash table $Q$ according to its keywords (lines 5–15). If the current traversal node is a non-leaf node, first prune according to the time and determine whether the Utime of the node intersects with the Utime of the query $q$. If there is an intersection, then there may be a POI in the child nodes of the node that meets the time familiarity. If the time attribute meets the requirement then the Bloom filter and keyword intersection are used to jointly determine whether its child nodes meet the query keyword requirement

and the child nodes that meet the requirement are entered into queue *U* (lines 16–27). If the current node is a leaf node, it operates the same as a non-leaf node when determining the time attribute. However, since the keyword information stored in the leaf node is of a string type, the KMP algorithm is used here to find whether there is a POI that meets the requirements (lines 28–39). When the queue *U* is empty, the result is returned as a hash table *Q* (line 40). The data stored in *Q* are <*key*, *o*> pairs, which are sorted in reverse order according to the zipper method based on the Score value when there is a storage conflict.

---

**Algorithm 2:** EKTIR-tree index-based filtering algorithm (Candidate algorithm)

---

**Input**: EKTIR-tree index on *D*, query *q* (*loc*, *K*+, *K*−, *st*, *et*).
**Output**: Candidate hash table *Q*.
begin
1:   Initialize the priority queue *U* and the hash table *Q* to empty; /* The index value of the priority queue is its score value */
2:   *U*.enqueue(KTIR-tree.root);
3:   while *U* is not empty then
4:     *e* ← *U*.dequeue; /* The head of queue *U* exits the queue */
5:     if *e* is POI then
6:       if *e.Utime* is included in *q.t* then
7:         for *key* ∈ *e.K* then
8:           if *key* ∈ *q.K*− then /* Exclusion keywords are skipped if present */
9:             continue; /* skip the current loop */
10:           *Q*.add (*e*); /* Positive keywords exist for the query, join *Q*
11:           end if
12:         end for
13:       else then
14:         continue; /* skip the current loop, Theorem 3 */
15:       end if
16:     else if *e* is a non-leaf node then
17:       if *e.Utime* intersects with *q.t* then
18:         for each keyword *key*− in *q.K*− then
19:           if *e.info.BloomFielter*(*key*−) is true && *key*-∈*e.Ukey* then
20:             continue; /* skip the current loop, Theorem 2 */
21:           end if
22:           if inverted file search for keyword records in *q.K*+ then
23:             *U*.enqueue(*e*); /* Theorem 1 */
24:           else then
25:             continue; /* skip the current loop, Theorem 3 */
26:           end if
27:       end for
28:     else then /* *e* is a leaf node */
29:       if *e.Utime* intersects with *q.t* then
30:         for each keyword *key*− in *q.K*− then
31:           if kmp(*e.CodeArr*, *Code*$_{key-}$)! = −1 then
32:             continue; /* skip the current loop */
33:           end if
34:           *U*.enqueue(*e*);
35:         end for
36:       end if
37:     end if
39:   end while
40:   return *Q*;
end

---

Assuming that the number of nodes in the EKTIR-tree index is *n*, the number of hash functions set by the Bloom filter is *m*, and *m* is a constant value, the time complexity of the Bloom filter query keyword in Algorithm 2 is O(*m*log*n*) and the time complexity of

the KMP algorithm used to query the leaf node keyword is O($M+N$). Since the average Huffman code length of keywords is $N$ = O(log$k$) in the worst case, assuming that the number of keywords of POIs in database $D$ is $S$ on average, then $M$ = O($S$log$k$) and the time complexity of searching the leaf part is O($S$log$k$). Therefore, the time complexity of Algorithm 2 is O($m$log$n$) + O($S$log$k$).

## 5. Spatial Keyword Group Query Method Based on Distance Domination Group

For the spatial keyword group query problem, this section discusses the underlying separation property based on the distribution law of spatial distance to POI and further proposes Properties 1 and 2 and Theorems 4–7 based on the separation property. In order to improve the efficiency of algorithmic queries, this section proposes Algorithms 3 and 4 based on the proposed properties and theorems.

Since this section only studies the properties and theorems on spatial distances, it will query the spatial coordinates of $q.loc$ and use $q$ to represent the coordinate points represented in the space.

**Separation Property** ([34]). *Suppose $M'$ is a feasible set and the distance cost between $M'$ and the query point $q$ is only related to the object $o_1$ in $M'$ that is farthest away from $q$ and the two objects $o_2$ and $o_3$ in $M'$ that are farthest away from each other.*

As shown in Figure 4, a query point $q$ and five space–text objects, $o_1$, $o_2$, $o_3$, $o_4$ and $o_5$, exist and the keyword attribute contents of all object points are visible in the figure. Suppose $q.K+ = \{k_1, k_2, k_5\}$, then there exists a feasible set as $M' = \{o_1, o_2, o_3\}$ with the distance dominator $o_1$ and diameter dominators $o_2$ and $o_3$ for $M'$. According to the separation property above, the distance cost of $M'$ is only related to these three objects within the set. We can calculate the spatial distance cost of the feasible set $M'$ by the formula in Definition 5. Here, it is considered that $o_1$, $o_2$ and $o_3$ form a distance domination group. Any feasible set with its query distance dominator as $o_1$ and diameter domination pairs as $o_2$ and $o_3$ is said to be ($o_1$, $o_2$, $o_3$) domination consistent and each ($o_1$, $o_2$, $o_3$) domination consistent feasible set has the same distance cost.

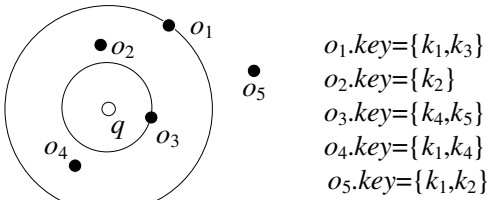

$o_1.key=\{k_1,k_3\}$
$o_2.key=\{k_2\}$
$o_3.key=\{k_4,k_5\}$
$o_4.key=\{k_1,k_4\}$
$o_5.key=\{k_1,k_2\}$

**Figure 4.** The figure of example objects.

A distance-dominated group-based approach: based on the separation property, we propose a distance-dominated group-based query approach. The method maintains a variable set S to store the currently found optimal feasible set and then iterates to find different feasible sets, taking the optimal solution each time until the final result is obtained. The main steps are:

(1)  Step 1: Find the distance dominator and select an object $o$ in the spatial–text object as the distance dominator of the current feasible set.
(2)  Step 2: Find the diameter dominating pair, select two objects $o_1$ and $o_2$ in the spatial–text objects as the distance dominating pair of the current feasible set and $o$, $o_1$ and $o_2$ can all cover $q.K+$ and form a feasible set $M'$.
(3)  Step 3: Find the suboptimal feasible set. Find the set $M$ for which ($o$, $o_1$, $o_2$) dominates consistently, and update $M'$ if it exists and Scost($M$) < Scost($M'$).
(4)  Step 4: Iterate, repeat steps 1 and 2, find another distance-dominating group and continue with step 3 until all dominating groups are traversed.

Steps 1–4 give a search strategy based on all possible distance-dominated groups, but the time complexity is $O(|N|^3)$, which is too inefficient. Therefore, to improve the search efficiency, Properties 1 and 2 are proposed for pruning.

**Property 1.** *Given a feasible set $M'$, if o is a distance dominator of $M'$, then the two objects of the diameter-dominating pair of $M'$ are also in the circular region with q as the center and dist(o, q) as the radius, i.e., C(q, dist(o, q)).*

**Proof of Property 1.** A distance dominator is the object in the feasible set that is farthest from the query point $q$, so for any object $o'$ in $M'$ other than $o$, we have $\text{dist}(o', q) \leq \text{dist}(o, q)$, i.e., $o'$ is in $C(q, \text{dist}(o, q))$. $\square$

**Property 2.** *Given a feasible set of $M'$ and o is a distance dominator of $M'$, with $o_2$ and $o_3$ being diameter-dominated pairs of $M'$, all possible objects of $M'$ are in the region $R = C(q, \text{dist}(o, q)) \cap C(o_1, \text{dist}(o_1, o_2)) \cap C(o_2, \text{dist}(o_1, o_2))$.*

**Proof of Property 2.** For every object, $o' \in M'$, there is $\text{dist}(o', q) \leq \text{dist}(o, q)$, i.e., $o'$ is in $C(q, \text{dist}(o, q))$. For each object, $o' \in M'$, there is $\text{dist}(o', o_1) \leq \text{dist}(o_1, o_2)$, i.e., $o'$ is in $C(o_1, \text{dist}(o_1, o_2))$, and $\text{dist}(o', o_2) \leq \text{dist}(o_1, o_2)$, i.e., $o'$ is in $C(o_2, \text{dist}(o_1, o_2))$. $\square$

Taking Figure 4 as an example, assuming here that $o_1$ has been chosen as the distance dominator and $o_2$ and $o_3$ as the diameter-dominating pair, there is still no need to consider $o_4$ as an object in the feasible set $M'$ because, firstly, $o_4$ violates the distance dominator property and, secondly, since $\text{dist}(o_2, o_4) > \text{dist}(o_2, o_3)$, $o_4$ violates the separation property when the diameter-dominating pair is $(o_2, o_3)$, so there is no need to consider $o_4$ either.

By Properties 1 and 2, the query method based on the distance-dominating group can reduce the search space in a large area. However, according to the usual situation, that the optimal set will be formed near the query point $q$, the distance nearest the first method is adopted in step 1 to determine the distance dominator. Taking Figure 3 as an example, when $o_3$ is selected as the distance dominator in step 1, no object exists in the delimited region $C(o_3, \text{dist}(o_3, q))$ in step 2, so that no diameter dominating pair can be found. In view of this phenomenon, Theorems 4 and 5 are given in this paper to find the nearest farthest distance dominator.

The set of nearest neighbors of $q$ is denoted by $N(q)$ and $r(C)$ denotes the radius of the circular region C.

**Theorem 4.** *Use rmin = max(dist(o, q) | o∈N(q)) when and only when rmin ≤≤ r(C), then a feasible set in region C exists.*

**Proof of Theorem 4.** Sufficiency is obvious because for $\forall C$, $r(C) \geq rmin$, all $N(q)$ in region C are a feasible set. Necessity is proved by contradiction. Suppose $r(C) < rmin$ and a feasible set $M$ exists in region C. Let $o_f$ be the object farthest from the query point in $N(q)$, i.e., $rmin = \text{dist}(o_f, q)$. If a keyword $key \in o_f.K \cap q.K+$ exists, an object that contains the keyword key and is closer to the query point $q$ than $o_f$ does not exist, otherwise $o_f \notin N(q)$. If $M$ is a feasible set, an object $o \in M$ exists and the set of keywords of o contains the key. Therefore, there is a conclusion $\text{dist}(o, q) \leq r(C) \leq rmin = \text{dist}(o_f, q)$, which obviously contradicts the original proposition. $\square$

Theorem 4 shows that if $r(C) \leq rmin$, then there is no feasible set in region C. So, the region of radius $rmin$ is the smallest region to be considered in this paper, the boundary objects in this region are the nearest distance dominators and the boundary objects are distributed along the region boundary.

**Theorem 5.** *Suppose M is a feasible set and rmax = Scost(M) and r(C) > rmax for region C. Then, for any feasible set $M'$ containing at least one object other than C, Scost(M') > Scost(M).*

**Proof of Theorem 5.** $Scost(M') \geq \max(dist(o, q) \mid o \in M') > r(C) > rmax = Scost(M)$, assuming that a feasible set $M'$ outside $M$ exists. Obviously the value of $Scost(M')$ is greater than the distance between the distance dominator of $M'$ and the query point q and this distance must be greater than r(C) of $M$, i.e., $Scost(M') > Scost(M)$. □

Theorem 5 shows that when a feasible set $M$ is known, there is no need to consider objects outside the region $C(q, rmax)$, where $rmax = Scost(M)$, and again, here, the objects are distributed along the region boundary.

Theorems 4 and 5 illustrate the minimum and maximum regions to be considered in this paper, specifically, the distance dominators o in the feasible set $M'$ must be distributed in a circle which is the region formed by subtracting the minimum region from the maximum region.

Let $M$ be a feasible set such that $rmin = \max(dist(o, q) \mid o \in N(q))$ and $rmax = Scost(M)$, defining the circular representation of the set $M$ as $Ring(M)$, i.e., $C(q, rmax) - C(q, rmin)$. As the radius of the outer circular region of $Ring(M)$ is $Scost(M)$, the region occupied by $Ring(M)$ becomes smaller as $Scost(M)$ becomes smaller. Based on the Properties 1 and 2 and $Ring(M)$, we further give Algorithm 3 to find the optimal set.

---

**Algorithm 3:** Finding the optimal set (FOPS algorithm)

---

**Input**: Query point $q$ and candidate hash table $Q$.
**Output**: The Optimal set $M$.
begin
1:    $M \leftarrow N(q)$; *Pairs* $\leftarrow \emptyset$;
2:    while $Ring(M)$ has unprocessed objects then
3:        $o \leftarrow$ the closest unprocessed object; /* step 1*/
4:        $C \leftarrow$ the region with $q$ as the center and radius of $dist(o, q)$; /* step2, Pairs are arranged internally in ascending order of distance between two points */
5:        *Pairs* $\leftarrow$ All possible diameter dominance pairs located within C;
6:        for each$(o_1, o_2) \in$ *Pairs* then
7:            if there exists a $(o, o_1, o_2)$-dominated consistent feasible set $M'$ in C **then**
8:                if $Scost(M) > Scost(M')$ then
9:                    $M \leftarrow M'$;
10:                    break;
11:                end if
12:        end for
13:        $o$ is marked as processed;
14:    end while
15:    return $M$;
end

---

As the initial data in the spatial–text database $D$ have been selected by Algorithm 2 for all POIs that meet the keyword and time requirements, only the optimal set needs to be selected from the candidate hash table $Q$ here. Algorithm 3 initializes the set *Pairs* to be empty and maintains a set $M$ to store the currently sought optimal set, initialized as the set of nearest neighbors of the query point, and then proceeds to the iterative process (line 1): The unprocessed objects in $Ring(M)$ are checked and the closest one is selected as the distance dominator of this feasible set in step 1. A circular region C with the query point as the center and $dist(q, o)$ as the radius is drawn in step 2. All possible diameter-dominated pairs are found in region C to add to the set *Pairs* (lines 2–5) and, in step 3, each pair of diameter-dominated pairs $(o_1, o_2)$ in the set *Pairs* are iterated to find if there is an existing set where $(o, o_1, o_2)$ is dominated consistently. If such a set $M'$ is found and the $Scost(M')$ is smaller than the cost of the currently known optimal set, the update of the optimal set is performed and then the traversal of the objects in the set *Pairs* is stopped (lines 6–15). If such a set $M'$ is not found, the traversal of the next pair of diameter-dominated pairs continues until it is found or all the objects in the *Pairs* are processed.

In order to further improve the efficiency of the algorithm, step 2 can refer to the results of the previous iteration process in different rounds of the iterative process to improve the efficiency of the screening process in this round. Assuming that the distance dominator selected in step 1 is $o$, construct region C and construct the set of *Pairs* in C. Then, in the next iteration, the distance dominator selected in the next round is $o'$ and the set is *Pairs'*. Since there are always *Pairs*∈*Pairs'*, another set *AddPairs* is constructed to hold the extra diameter-dominating pairs in *Pairs'* relative to *Pairs*, i.e., *AddPairs* = {($o'$, $o''$) | $o''$∈C($q$, dist($o$, $q$))}, then *Pairs'* = *Pairs*∪*AddPairs*. Moreover, it is not necessary to add all possible diameter dominators to the set of *Pairs* in step 2. Theorems 6 and 7 are given here to effectively improve the pruning efficiency of the candidate diameter-domination pairs.

**Theorem 6.** *Suppose M is a feasible set with distance dominator o and diameter-dominated pair* $(o_1,o_2)$*, then* $dist(o_1,o_2) > dist(o, q) - min\{dist(o_1, q), dist(o_2, q)\}$*.*

**Proof of Theorem 6.** It is easy to know that $dist(o_1, o_2) > dist(o_1, o)$ and $dist(o_1, o_2) > dist(o_2, o)$. Based on the trigonometric inequality, it is known that $dist(o_1, o) + dist(o_1, q) > dist(o, q)$ and $dist(o_1, o) + dist(o_2, q) > dist(o, q)$, which can be derived from $dist(o_1, o_2) > dist(o, q) - min\{dist(o_1, q), dist(o_2, q)\}$, so it is known that there is an existing minimum boundary value for the diameter-dominated pair $(o_1, o_2)$. □

**Theorem 7.** *Suppose M is a feasible set with distance dominator o and diameter dominating pair* $(o_1,o_2)$ *such that M' is another feasible set. Scost(M) ≤ Scost(M') when, and only when,* $dist(o_1,o_2)$ *≤ Scost(M') − dist(o, q).*

**Proof of Theorem 7.** Proof by contradiction, assume that $Scost(M) \geq Scost(M')$, i.e., $dist(o, q) + dist(o_1, o_2) \geq Scost(M')$, which introduces $dist(o_1, o_2) \geq Scost(M') - dist(o, q)$, contradicting the original condition. □

Let $M'$ be the feasible set found so far, Theorems 6 and 7 show that the diameter-dominated pair $(o_1, o_2)$ can be pruned if $dist(o_1, o_2) > Scost(M') - dist(o, q)$. Suppose $dmax = Scost(M') - dist(o, q)$, then $dmax$ is the maximum boundary value of the diameter-dominated pair $(o_1, o_2)$. In summary, it is only necessary to select the diameter-dominated pair between the minimum and maximum boundary values in step 2.

Step 3 finds the feasible set for which $(o, o_1, o_2)$ dominates consistently, which involves the processing of keywords, and Algorithm 4 shows the relevant computational logic details in detail.

In Algorithm 4, a feasible set with the consistent domination of $(o, o_1, o_2)$ is found then its output is direct, otherwise the output is ∅. First of all, one must determine whether dist $(o_1, o_2) \geq max\{dist (o, o_1), dist (o, o_2)\}$ is true based on the nature of the diameter branch pair in the Separation Property (line 1). If not, no feasible set is uniformly dominated by $(o, o_1, o_2)$. Directly output the result until the algorithm ends; otherwise proceed to the next step of judgment. Initialize $M'$ to $\{o, o_1, o_2\}$ and also maintain a set (*unkey*) to save the keywords that currently need to be searched (lines 2–3). If the *unkey* is ∅, it indicates that the currently found object set has covered all the query keywords and we can directly output $M'$ (lines 4–5). Otherwise, we need to iterate through the POIs related to the keywords in the *unkey* and find the objects that can form a consistent set of dominators feasible with the currently existing objects. First, after reducing the search space to R by Property 2, the traversal can be based on the hash table $Q$ output by Algorithm 2 to select the best object in R for the desired keyword (lines 6–13). Then, we add the objects matching the keyword to $M'$ and check whether $M'$ is dominated consistently, i.e., check whether $o_1$ and $o_2$ are still the diameter-dominated pair of the set. If yes, output $M'$ (lines 14–18), otherwise restore $M'$ and check the next subset of objects. If at the end, there is still no feasible set found, then output ∅ (lines 19–24).

---

**Algorithm 4:** Find the feasible set $M'$ for which $(o, o_1, o_2)$ dominates consistently

---

**Input**: Three spatial text objects $o, o_1, o_2$.
**Output**: If it exists then output $(o, o_1, o_2)$ dominates the consistent feasible set $M'$, otherwise output $\varnothing$.
begin
1:  if dist$(o_1, o_2) \geq$ max{dist$(o, o_1)$,dist$(o, o_2)$} then
2:      $M' \leftarrow \{o, o_1, o_2\}$;
3:      $unkey \leftarrow q.K+ - (q.K+ \cap (o.K \cup o_1.K \cup o_2.K)$; /* keywords not yet collected */
4:      if $unkey$ is $\varnothing$ then
5:          return $M'$;
6:      else then
7:          R $\leftarrow$ C$(q$, dist$(o, q)) \cap$C$(o_1$, dist$(o_1, o_2)) \cap$C$(o_2$, dist$(o_1, o_2))$; /* Property 2 */
8:          find all object points in $R$ with keywords related to $unkey$ from $Q$ into the middle set *Temp*
9:          if the keywords of the objects in *Temp* cannot be overwritten together with $unkey$ then
10:             return $\varnothing$;
11:         else then
12:             for each *curkey* in $unkey$ then
13:                 $e \leftarrow Q$.get(*curkey*);
14:                 if $e \in$ *Temp* then
15:                 $M' \leftarrow M' \cup \{e\}$; /* The best POI corresponding to each desired keyword from the hash table $Q$ and which is within R is added to the set $M'$ */
16:                 else then
17:                     if $M'$ is $(o, o_1, o_2)$ dominated consistently then
18:                         return $M'$;
19:                     else then
20:                         $M' \leftarrow M' - e$; /* reset $M'$ */
21:                     end if
22:             end for
23:         end if
24:     end if
25: end if
26: return $\varnothing$;
end

---

In Algorithm 4, assuming that $|q.K+|$ is $u$ and $|o.K|$ is $v$ on average, the time complexity of the set operation used to compute the *unkey* is O$(u + v)$ and the time complexity of searching for keyword-related POIs from the hash table is O(1). However, the time complexity of determining whether the keywords of the selected objects cover the *unkey* as a whole is O$(v)$, so the time complexity here is O$(v)$, and then calculating whether the relevant objects in R can be formed into a feasible set, and the time complexity of this part is O$(u)$. Therefore, the overall time complexity of Algorithm 4 is O$(u + v)$.

Since Algorithm 4 forms part of Algorithm 3, the time complexity of Algorithm 3 can be analyzed only after the time complexity analysis of Algorithm 4 is complete. In Algorithm 3, assuming that the number of POIs in the database $D$ is $p$, the time complexity of finding N$(q)$ is O(log$p$), the time complexity of finding the distance-dominant and diameter-dominant pairs and ranking them is O$(p$log$p)$ and the time complexity of finding the suboptimal feasible set in Step 3 is O$(p^2u + p^2v)$, so that Algorithm 3 has a time complexity of O$(p$log$p)$ + O$(p^2u + p^2v)$.

Up to this point, the optimal set of all POI objects can be obtained after two stages of Algorithms 2 and 3, so this paper proposes a query method based on the EKTIR-tree index and dominating group (EKTDG). The method consists of two parts, the first part is based on the EKTIR-tree index using Algorithm 2 to perform preliminary pruning and filtering operations on POI objects to get the candidate objects that meet the time and keyword requirements and then Algorithm 3 is used to refine the candidate objects to select the result set with the optimal distance.

### 6. Experiment Analysis

For the problem of a time-aware group query with exclusion keywords, this paper proposes a query method based on an EKTIR-tree index and dominating groups. The proposed method first performs pruning queries on temporal and keyword attributes based on the established EKTIR-tree index and filters out all POIs in the spatial–text database *D* that meet the temporal and keyword requirements into the hash table. Further based on the information of the hash table, the final optimal set is found based on the spatial distance and domination group. In order to evaluate the method performance, four aspects of comparison experiments are designed in this section. The first aspect compares the effect of the dataset size on the efficiency of different algorithms, the second aspect compares the effect of the number of query's positive keywords on the efficiency of different algorithms, the third aspect compares the effect of the number of query rejection keywords on the efficiency of different algorithms and the fourth aspect compares the accuracy of the execution result$\frac{3}{4}$s of different algorithms on different datasets. The methods that are compared with the algorithm proposed in this paper are the CD-Exact algorithm [27], the Unified-E algorithm [14] and the EXACT algorithm [16].

The environment used for the experiments is the Microsoft Windows 10 (64-bit), Core(TM) i7-7500U CPU@2.70 GHz processor, with a running memory of 12 GB, and the programming language is Java1.8.

The experimental data were obtained from the real data set with small data processing to make it more suitable for the experimental needs. In this paper, we use three datasets, Yelp, Hotel and GN, where Yelp is downloaded from the Yelp US dataset and each POI has a location coordinate and a set of classification tags, which can be considered as a set of keywords. The dataset Hotel contains information about some hotels in the United States and each POI has unique location information and a set of keywords to describe the characteristics of the hotel. The dataset GN is from the U.S. Geographic Names Committee (geonames.usgs.gov), where each object has a location and a set of descriptive keywords (e.g., a place name, such as valley). And [0–24] time information is randomly generated for each object in the above dataset. Table 1 shows the information table of the dataset used in the experiment.

**Table 1.** Datasets used in the experiments.

| Dataset | GN | Hotel | Yelp |
|---|---|---|---|
| Number of objects | 1,868,821 | 20,790 | 192,609 |
| Number of unique words | 222,409 | 602 | 2468 |
| Number of words | 18,374,228 | 80,645 | 788,841 |

**Experiment 1.** *This part of the experiment aims to compare the efficiency of the EKTDG algorithm with the CD-Exact algorithm, Unified-E algorithm and EXACT algorithm in terms of dataset size. Specifically, each dataset is randomly selected in 2–12 M size increments with a tolerance of 2 for comparison experiments and all other conditions are controlled equally. As the amount of data increases, the CPU execution time of the four algorithms changes, as shown in Figure 5.*

Figure 5 shows that the CPU execution time of the EKTDG algorithm does not increase steeply as the amount of data increases and the scalability for the data set performs better compared to other comparative algorithms because the EXACT and Unified-E algorithms do not use indexing techniques and have fewer pruning methods. When the volume of data increases, the query time increases dramatically by relying only on the spatial distance pruning method. However, the EKTDG method proposed in this paper first uses indexing techniques to efficiently filter keywords and temporal information to narrow the search space and then prunes the search object in the space after the previous narrowing step to query the results; most POIs are efficiently filtered out by the index in the first step, so the algorithm has good scalability.

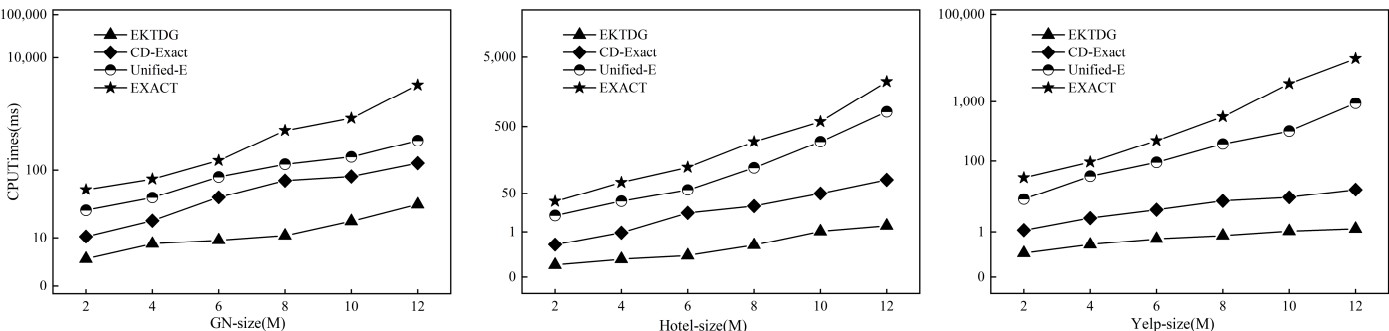

**Figure 5.** The effect of dataset size on algorithm efficiency.

**Experiment 2.** *This part of the experiment aims to compare the effect of a different number of query forward keywords on the efficiency of various algorithms. Specifically, for each dataset, a certain number of keywords are randomly generated from all the keyword information of POI in this dataset as query positive keywords whose number variation interval is [1–5]. The CPU execution time variation of the four algorithms is shown in Figure 6.*

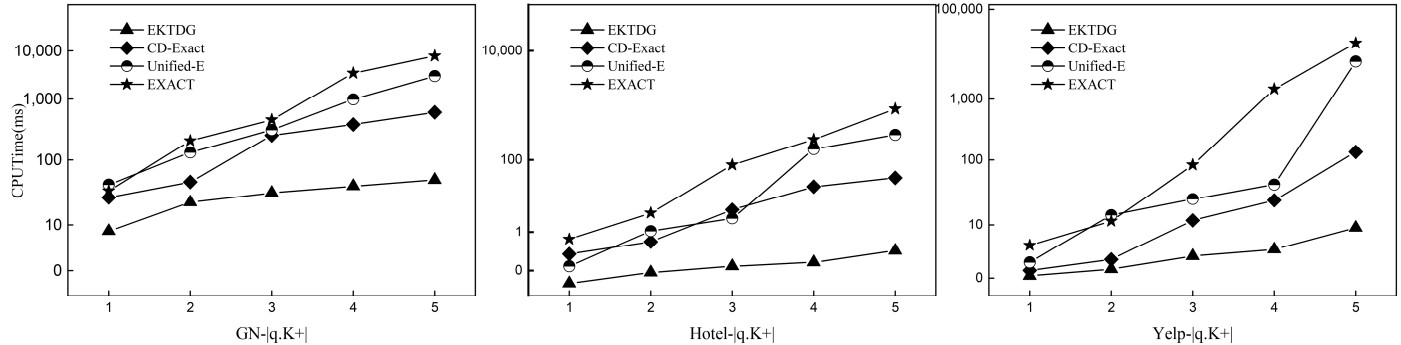

**Figure 6.** The effect of positive keywords number on algorithm efficiency.

As can be seen from Figure 6, the EKTDG algorithm does not change significantly and is more efficient than the other algorithms as the number of query positive keywords increases. This is because the EKTDG algorithm introduces the idea of Huffman encoding on the index, which improves the hit rate of query keywords to a certain extent and is more optimized compared to CD-Exact. The CD-Exact algorithm also uses the preprocessing of keywords to calculate the frequency of keywords to improve the efficiency of the algorithm, so it is more efficient overall than the Unified-E algorithm and the EXACT algorithm. The Unified-E algorithm uses pruning methods, while the EXACT algorithm is the least efficient because of the error in finding the exact result based on the approximation algorithm.

**Experiment 3.** *This part of the experiment aims to compare the effect of a different number of query exclusion keywords on the efficiency of various algorithms. Specifically, for each dataset, a certain number of keywords are randomly generated from all of the keyword information of POI in this dataset as exclusion keywords and their number varies in the interval [1–5]. For the other algorithms that do not consider the exclusion keywords, a keyword dichotomous tree approach, commonly used in the field, is added to them. The CPU execution time variation of the four algorithms is shown in Figure 7.*

As can be seen from Figure 7, there is little difference in the execution efficiency between the algorithms after adding the processing of the exclusion keyword to the three compared algorithms. When the exclusion keyword changes in the next query, the binary tree method needs to reconstruct the binary tree index and all the spatial indexes for the exclusion keyword, so the running time of this method grows faster. However, the EKTDG

algorithm can be created once and used continuously with a low maintenance cost. The algorithm introduces Bloom filters in the index, which are constructed for the keyword information of the whole database space. Here, not only is the query speed fast, but the space occupation is also small and subsequent maintenance is not required. Since the EKTDG algorithm first prunes the space based on the exclusion keywords, its running time slowly decreases as the number of exclusion keywords increases and the number of result sets is smaller. Therefore, the running time of the EKTDG algorithm grows slowly and runs faster as the number of excluded keywords increases.

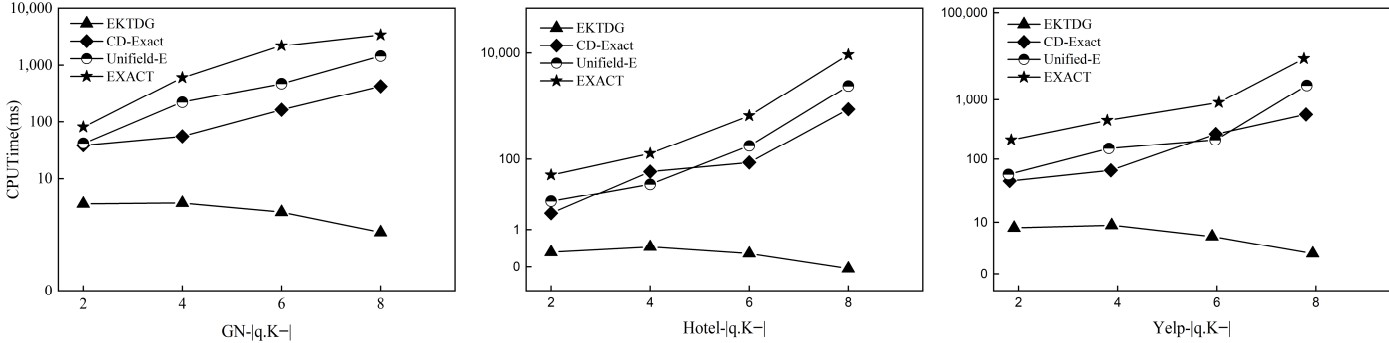

**Figure 7.** The effect of exclusion keywords number on algorithm efficiency.

**Experiment 4.** *This part of the experiment aims to compare the query accuracy of different algorithms on three datasets. Specifically, for each dataset, the four algorithms are applied with the same number of control query keywords and the same other metrics. The accuracy of algorithm execution is shown in Figure 8.*

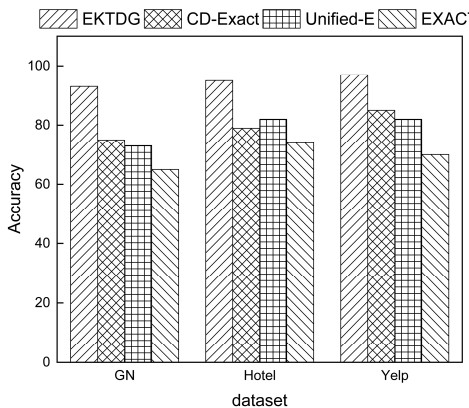

**Figure 8.** Algorithm accuracy.

As can be seen from Figure 8, the accuracy of the EKTDG algorithm execution can reach more than 90% for either dataset. This is due to the fact that the algorithm uses a backward index and Bloom filter technique that would have allowed for the accurate querying of keywords, but the Bloom filter can feature false positives resulting in a less than 100% accuracy. In contrast, CD-Exact and Unified-E can achieve about 80% accuracy, while the EXACT algorithm can only reach about 75% accuracy due to the large error caused by using an approximation algorithm to reduce the search space in the early stage.

## 7. Conclusions

For the traditional spatial keyword group query that does not take into account the temporal information and exclusion intention proposed by users, this paper proposes a new query problem, i.e., a time-aware group query problem with exclusion keywords. In

order to solve the problem efficiently, this paper proposes a query method based on an EKTIR-tree index and dominating group. The experimental results show that the algorithm proposed in this paper has good scalability and efficiency. Future research work will focus on the following aspects:

1. A study of spatial keyword group queries in road network environments.
2. A study of spatial keyword group queries under privacy protection.
3. A study of spatial keyword group queries in a dynamic environment with a streaming data style.

**Author Contributions:** Conceptualization, Liping Zhang and Jing Li; methodology, Liping Zhang and Jing Li; investigation, Liping Zhang; writing—original draft preparation, Jing Li; writing—review and editing, Jing Li, Liping Zhang and Song Li; project administration, Song Li. All authors have read and agreed to the published version of the manuscript.

**Funding:** This research was funded by the National Natural Science Foundation of China, grant number 62072136, the Natural Science Foundation of Heilongjiang Province, grant number LH2023F031, and the National Key R&D Program of China, grant number 2020YFB1710200.

**Data Availability Statement:** The data presented in this study are available on request from the corresponding author.

**Conflicts of Interest:** The authors declare no conflict of interest.

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
