# Peer review of "Research on Time-Aware Group Query Method with Exclusion Keywords"

_ijgi, doi:10.3390/ijgi12100438_

Round 1

Reviewer 1 Report

1) “exclusive keywords ” appeared in this paper should be corrected by “exclusion keywords”

2) In section 1, “Compared with the previous methods, the operation efficiency of the proposed 79 method is improved by 14%”, ____This improvement is achieved with respect to which previous algorithm, should be specified.

3) In section3, “exclusive keywords” in Definition1 should be corrected by “exclusion keywords”. 

In Definition2(1) , MS should be shown as M.

In Definition5, M.cost should be shown as M.Scost.

4) In Definiton8, the TF-IDF method should be indexed by its reference.

5) In theorem3, its proof should be given. Their design ideas of Equation (4) and Equation (5) should be analyzed and should be illustrated by some examples.

Reviewer 2 Report

It is an interesting article about time-aware group query method with exclusion keywords. The authors propose a query method based on the EKTIR-Tree index and dominating group (EKTDG).  The Candidate algorithm is proposed based on the EKTIR-tree index to filter out the spatial-textual objects that meet the query keywords and time requirements, narrowing the search space for subsequent queries on a large scale. To address the problem of low efficiency of existing algorithms based on spatial distance query, a distance dominating group is defined and a pruning algorithm based on spatial distance-dominating group is proposed. The theory and technology of this paper are excellent.

 Although the contribution is obvious, this paper has the following problems:

1.  Some sentences in the paper are redundant and not concise enough.

2.  It is suggested to provide an example to explain Definition 1.

3.  Please explain the function q.(st,et) in Definition 2.

4.  In algorithm 1, the initial value of HT should be an empty set symbol.

5.  In the proof of Theorem 2, Why the bit of key after hash function mapping must be 1 instead of 0? Please provide further explanation.

6.  In the explanation of Algorithm 2,the sentence “If the current traversal node is a non-leaf node, it is first pruned according to the time. is not accurate. Please explain in detail how to prune according to time.

7. In the section “Spatial keyword group query method based on distance domination group’, about the Step 1: Find the distance dominator, select an object o in the spatial text object as the distance dominator of the current feasible set. Can the author explain that object o is arbitrarily selected?

Moderate editing of English language required

Reviewer 3 Report

The time aware group Query method with exclusion keywords plays an important role in fields such as geographic information systems, spatial databases, and location services.The authors proposed an effective time-aware group query method with exclusion keywords. Firstly, the EKTIR-tree index is constructed for the POI objects in the space. Then the dominating group is selected among the candidate objects to determine the optimal solution by iterating continuously through the pruning method. The experimental results show that the algorithm proposed in this paper has good scalability and efficiency.The contribution of this paper is positive. The research work of this paper is significant. The content that needs further improvement is as follows:

1)In Section Introduction, the motivation and background of the research need to be further elaborated.

2) Please consider whether Algorithm 5 in this paper is necessary and whether it can be replaced by a textual description?

  3) The paper presents some important algorithms, and the quality of some sentences in the algorithms needs to be further improved. For example:The statement 'continue' in step 9, 13, 20, and 32 of algorithm 2 should be replaced by a clear processing statement. Does Pairs have an initial value in Algorithm 3? The author needs to clarify the initial value of Pairs.The statement 'else then' should be added after step 6, 11, and 16 of Algorithm 4, respectively.

  4) The relationship between the algorithms in the paper needs to be further discussed.

1) Some language and English writing are in need of further improvement.

Reviewer 4 Report

The paper studies time-aware group query problem with exclusive keywords. They propose an index structure called EKTIR-tree that incorporates Huffman coding and Bloom filters to increase the hit rate of keyword queries. A distance dominating group as well as a pruning algorithm are developed to efficiently answer the query. They perform the theoretical analysis and the experimental evaluation. The results demonstrate the performance advantage of the proposal.

Although the studied issue does not receive too much attention in the literature, the motivation needs to be improved. A concrete application example is preferred. There are some technique issues in terms of novelty and significance. Detailed comments are in the following.

D1. The motivation of considering temporal constraints is not well explained. For spatial queries/objects, there is no temporal parameter. What is application task/requirement of considering temporal constraints on spatial objects?

D2. Since a spatial object is considered, the meaning of start and end time should be explained. How about this method. One first finds all objects containing the query keywords and then iteratively filter those containing negative keywords.

D3. In Definition 1, one should formalize the query by expressions instead of words. For example, what is the condition of putting an object into the result set.

D4. What is the difference between the proposed structure and IR-tree? How does the structure manage the temporal data? How to perform temporal pruning based on the index?

D5. Algorithm 1 is similar to the procedure of building a Huffman tree, which is not quite novel. The output of Algorithm 4 does not look formal (actually the result is a set but could be an empty set). Algorithm 5 is too simple and can be ignored. Algorithm complexities should be analyzed.

D6. The purpose of distance dominating group is not clear. This could be utilized to find close objects but they may contain exclusive keywords.

D7. Some explanation about alternative methods are required. In Figure 4, the largest value on x-axis is 14k which is much smaller than the overall number of objects for all datasets. I expect that for the effect of exclusive keywords, the larger the value k is, the less time is required. This is because the size of the result set should be small for a large k.

D8. Is the proposed algorithm exact or approximate?

D9. English needs to be improved, e.g., “Given a time-aware spatial keyword group query with exclusive key- 164 words (TEGSKQ) is denoted as”.

Round 2

Reviewer 4 Report

The authors have carefully done the revision and the revision is better than the original one. A few comments are provided in the following. 

 D1. Although the authors explain the difference between the IR-tree and the EKTIR-tree, the novelty is not significant. How to update the extended information in the EKTIR-tree?

D2. Figure 1 does not look nice. I expect that one should provide a picture providing the relationship among subroutines. Another point is that 14% performance improvement is not quite significant.

D3. The way of presenting algorithms is not formal, e.g., “jump out of the current loop”.

D4. There is still a number of English issues such as “Definition(s) 7-9.” Please check carefully.

The authors have carefully revised the manuscript. The revision is better than the original one. A few comments are provided in the following.

D1. Although the authors explain the difference between the IR-tree and the EKTIR-tree, the novelty is not significant. How to update the extended information in the EKTIR-tree?

D2. Figure 1 does not look nice. I expect that one should provide a picture providing the relationship among subroutines. Another point is that 14% performance improvement is not quite significant.

D3. The way of presenting algorithms is not formal, e.g., “jump out of the current loop”.

D4. There is still a number of English issues such as “Definition(s) 7-9.” Please check carefully.
